# Comparison of Liquid-Based Preparations with Conventional Smears in Thyroid Fine-Needle Aspirates: A Systematic Review and Meta-Analysis

**DOI:** 10.3390/cancers16040751

**Published:** 2024-02-11

**Authors:** Yun Jin Kang, Hyeon Woo Lee, Gulnaz Stybayeva, Se Hwan Hwang

**Affiliations:** 1Department of Otorhinolaryngology-Head and Neck Surgery, Soonchunhyang University College of Medicine, Cheonan 14584, Republic of Korea; savie87@gmail.com (Y.J.K.); fawa7878@naver.com (H.W.L.); 2Department of Physiology and Biomedical Engineering, Mayo Clinic, Rochester, MN 55905, USA; stybayeva.gulnaz@mayo.edu; 3Department of Otolaryngology-Head and Neck Surgery, Bucheon St. Mary’s Hospital, College of Medicine, The Catholic University of Korea, Seoul 06591, Republic of Korea

**Keywords:** thyroid gland, fine-needle aspiration biopsy, cytology, conventional smears, liquid-based preparation, meta-analysis

## Abstract

**Simple Summary:**

We compared the diagnostic accuracy of conventional smears and liquid-based preparations for detecting thyroid lesions using fine-needle aspiration cytology. We reviewed 15,861 samples from 17 studies. There was no significant difference between conventional smears and liquid-based preparations in terms of diagnostic accuracy or the proportion of inadequate smears. SurePath outperformed ThinPrep in terms of diagnostic accuracy among the liquid-based preparations. Recommendations for one method over another should take cost, feasibility, and accuracy into account, necessitating additional research.

**Abstract:**

Background: To compare conventional smears (CSs) and liquid-based preparations (LBPs) for diagnosing thyroid malignant or suspicious lesions. Methods: Studies in the PubMed, SCOPUS, Embase, Web of Science, and Cochrane database published up to December 2023. We reviewed 17 studies, including 15,861 samples. Results: The diagnostic odds ratio (DOR) for CS was 23.6674. The area under the summary receiver operating characteristic curve (AUC) was 0.879, with sensitivity, specificity, negative predictive value, and positive predictive value of 0.8266, 0.8668, 0.8969, and 0.7841, respectively. The rate of inadequate specimens was 0.1280. For LBP, the DOR was 25.3587, with an AUC of 0.865. The sensitivity, specificity, negative predictive value, and positive predictive value were 0.8190, 0.8833, 0.8515, and 0.8562. The rate of inadequate specimens was 0.1729. For CS plus LBP, the AUC was 0.813, with a lower DOR of 9.4557 compared to individual methods. Diagnostic accuracy did not significantly differ among CS, LBP, and CS plus LBP. Subgroup analysis was used to compare ThinPrep and SurePath. The DORs were 29.1494 and 19.7734. SurePath had a significantly higher AUC. Conclusions: There was no significant difference in diagnostic accuracy or proportion of inadequate smears between CS and LBP. SurePath demonstrated higher diagnostic accuracy than ThinPrep. Recommendations for fine-needle aspiration cytology should consider cost, feasibility, and accuracy.

## 1. Introduction

Thyroid nodules are predominantly benign but exhibit a prevalence of ~4–7% in the general population [1]. Papillary thyroid carcinoma, the most frequent among malignant lesions, has been increasing in prevalence [2,3]. Due to the extensive vascularization of the thyroid, histological biopsies are challenging to perform. Consequently, fine-needle aspiration cytology (FNAC) has become the primary minimally invasive diagnostic method for these nodules [4,5,6].

Conventional smear (CS) is a common method of FNAC for thyroid lesions [7]. It is recognized for its simplicity and convenience [8]. Furthermore, it is relatively safe, repeatable, and low risk [6,7,9,10]. However, CS can report variable results, depending on the uneven thyroid tissue samples or cytopathologist’s experience [8,10,11]. Artifacts may also arise during the drying of specimens, and results can vary by technician [9]. In addition, the presence of fibrosis and cystic lesions can result in poor cellularity [8]. These limitations can lead to a ~50% increase in inadequate specimens, complicating accurate diagnoses by pathologists [12].

The liquid-based preparation (LBP) method is a novel diagnostic approach in FNAC and is extensively used for breast and salivary gland examinations [1,13,14]. Introduced in 1996 as an alternative to the traditional Papanicolaou smear, LBP aims to standardize samples by minimizing artifacts and errors inherent in CS [1,8,10,15]. Two commonly used kits are ThinPrep (Hologic, Marlborough, MA, USA) and SurePath (BD Diagnostics-TriPath Imaging, Burlington, NC, USA). LBP involves collecting aspirates in a special fixative and employing an automated machine to reduce cell debris, inflammatory cells, red blood cells, and artifacts, thus producing a uniformly distributed Papanicolaou smear slide [8,9]. Through processes such as homogenization, vacuum application, and sedimentation, it provides well-preserved sample cells against a clean background [8,10].

However, the effectiveness of LBP for diagnosing thyroid lesions, where cell cluster shape and background are crucial, remains debatable [8]. Few studies and reviews have compared CS and LBP, and many exhibit a bias toward LBP, with varied criteria for evaluating sensitivity and specificity [8,10,13,14,16,17,18]. We performed a comparative meta-analysis of the diagnostic accuracy and rate of inadequate smears (RISs) between CS and LBP in FNAC of malignant or suspicious thyroid lesions, incorporating the latest research. In addition, we conducted subgroup analyses comparing two common LBP kits, ThinPrep and SurePath.

## 2. Materials and Methods

### 2.1. Study Protocol and Registration

This systematic review and meta-analysis followed the Preferred Reporting Items for Systematic Reviews and Meta-Analyses guidelines [19] and was conducted in accordance with recommendations for optimal searches of the literature in systematic reviews within the field of surgery [20]. The study protocol was prospectively registered on the Open Science Framework (https://osf.io/zj4hv/, accessed on 11 December 2023).

### 2.2. Literature Search Strategy

Clinical studies were sourced from PubMed, SCOPUS, Embase, Web of Science, and the Cochrane Central Register of Controlled Trials up to December 2023. The search terms included ‘thyroid gland’, ‘fine-needle aspiration’, ‘fine-needle aspiration biopsy’, ‘cytology’, ‘cytopathology’, ‘conventional smear’, ‘direct smear’, and ‘liquid-based preparation’. We also reviewed the references of identified articles to ensure no relevant studies were overlooked. Two independent reviewers scrutinized all abstracts and titles for eligible studies, excluding those unrelated to the diagnosis of thyroid malignancies or suspicious lesions through cytologic examination based on fine-needle aspiration and confirmed by surgical histologic examination.

### 2.3. Selection Criteria

The inclusion criteria were patients undergoing fine-needle aspiration biopsy for thyroid lesions, prospective or retrospective studies, studies comparing the diagnostic accuracy of CS and LBP against surgical histologic findings, and the availability of data for sensitivity and specificity analysis. The exclusion criteria included case reports, review articles, studies on other head and neck lesions such as neck lymph nodes or neck masses, and data not applicable for assessing the diagnostic value of imaging studies. The search strategy was summarized in a flow diagram to screen studies selected for the meta-analysis (Figure 1).

### 2.4. Data Extraction and Risk of Bias Assessment

Among all studies included in the meta-analysis, studies published after 2010 evaluated thyroid malignant lesions using the Bethesda system of reporting thyroid cytopathology. Other included studies that did not use the Bethesda system of reporting thyroid cytopathology were also compared, including suspicions for malignancy, malignant, nondiagnostic, or inadequate lesions evaluated by the Bethesda system. Because the cytologic categories, except malignant lesions, were different for each included study, it was difficult to evaluate benign and atypia lesions. Therefore, it was difficult to expect malignancy risk in the different diagnostic classes, and we summarized the numbers of excluded categories (Appendix A).

Data were abstracted using standardized forms by two independent reviewers [21]. The outcomes for analysis included diagnostic accuracy (diagnostic odds ratio (DOR)), the summary receiver operating characteristic (sROC) curve, and the area under the curve (AUC) [6,9,17,22,23,24,25,26,27,28,29,30,31,32,33,34,35,36,37,38,39,40,41].

The DOR was calculated as (true positive (TP)/false positive (FP))/(false negative (FN)/true negative (TN)) to assess diagnostic accuracy with 95% confidence intervals (CIs) using random-effects models that accounted for both within- and between-study variation. The DOR values range from 0 to infinity, with higher values indicating better diagnostic performance. A value of 1 suggests that the test provides no diagnostic advantage. The sROC is preferred for meta-analyses of studies reporting sensitivity and specificity pairs. As the discriminatory power of a test increases, the sROC curve approaches the top left corner in the ROC space, where sensitivity and specificity both equal 1 (100%) [42]. The AUC, ranging between 0 and 1, reflects test performance quality; values between 0.90 and 1.0 are considered excellent, 0.80–0.90 are good, 0.70–0.80 are fair, 0.60–0.70 are poor, and 0.50–0.60 are considered failures [43].

Data extracted from the studies included the number of patients, the correlations among scores in endoscopy and computed tomography, and TP, TN, FP, and FN for AUC and DOR calculations. The Quality Assessment of Diagnostic Accuracy Studies version 2 tool was employed to evaluate methodological quality and risk of bias [44].

### 2.5. Statistical Analysis and Outcome Measurements

The ‘R’ statistical software (Version 4.3.2) (R Foundation for Statistical Computing, Vienna, Austria) was used for meta-analysis. Homogeneity analyses employed the Q statistic to assess heterogeneity. Subgroup analyses were conducted using different types of imaging studies. Forest plots were used to depict sensitivity, specificity, and sROC curves. Begg’s funnel plot and Egger’s linear regression test were performed to evaluate potential publication bias.

## 3. Results

### 3.1. Search and Study Selection

In total, 17 studies, including 15,861 samples, were included in the analysis (Figure 1). The characteristics of the studies are detailed in Table 1, and the bias assessment results are given in Table 2. Egger’s test was significant (*p* < 0.05), indicating no apparent bias in the included studies, as suggested by Begg’s funnel plot (Figure 2).

### 3.2. Diagnostic Accuracy

In the case of CS, the DOR was 23.6674 (95% CI [13.4718; 41.5794]; I^2^ = 82.6%) (Figure 3). The AUC was 0.879 (Figure 4). Sensitivity, specificity, and negative predictive value of CS were 0.8266 (95% CI [0.7498; 0.8835]; I^2^ = 87.2%), 0.8668 (95% CI [0.7721; 0.9259]; I^2^ = 96.1%), and 0.8969 (95% CI [0.7805; 0.9552]; I^2^ = 97.6%), respectively. The RIS was 0.1280 (95% CI [0.0865; 0.1853]; I^2^ = 98.5%).

For all LBP methods combined, the DOR was 25.3587 (95% CI [7.1871; 89.4747]; I^2^ = 95.9%) (Figure 3). The AUC was 0.865 (Figure 4). Sensitivity, specificity, and negative predictive value were 0.8190 (95% CI [0.7459; 0.8746]; I^2^ = 83.8%), 0.8833 (95% CI [0.7348; 0.9539]; I^2^ = 97.8%), and 0.8515 (95% CI [0.7124; 0.9300]; I^2^ = 95.6%), respectively. The RIS was 0.1729 (95% CI [0.1231; 0.2375]; I^2^ = 97.0%).

When combining CS with LBP, the DOR was 9.4557 (95% CI [3.2976; 27.1139]; I^2^ = 93.6%) (Figure 3). The AUC was 0.813 (Figure 4). Sensitivity, specificity, and negative predictive value were 0.7809 (95% CI [0.6976; 0.8463]; I^2^ = 79.6%), 0.7267 (95% CI [0.5370; 0.8591]; I^2^ = 95.8%), and 0.8111 (95% CI [0.6388; 0.9125]; I^2^ = 95.2%), respectively. The RIS was 0.1109 (95% CI [0.0901; 0.1357]; I^2^ = 82.1%).

While there were no statistically significant differences in diagnostic accuracy and RIS among CS, LBP, and their combination, CS plus LBP appeared to have a relatively lower diagnostic accuracy compared to CS and LBP individually. Conversely, the combination of CS with LBP tended to reduce the RIS (Appendix A).

### 3.3. Subgroup Analysis of Diagnostic Accuracy According to the Methods of LBP

Several LBP kits were included in the enrolled comparative studies, including ThinPrep, SurePath, CellPrepPlus, and an unspecified tool. Among these, ThinPrep and SurePath are the most commonly used. A subgroup analysis was conducted to determine which method is more accurate for diagnosing thyroid malignancies or suspicious lesions.

For ThinPrep, the DOR was 29.1494 (95% CI [4.9108; 173.0254]; I^2^ = 89.6%). The AUC was 0.791. Sensitivity, specificity, negative predictive value, and positive predictive value were 0.8182 (95% CI [0.7403; 0.8767]; I^2^ = 8.7%), 0.9080 (95% CI [0.7064; 0.9759]; I^2^ = 94.7%), 0.8988 (95% CI [0.5741; 0.9832]; I^2^ = 96.4%), and 0.7989 (95% CI [0.3465; 0.9675]; I^2^ = 96.7%), respectively.

For SurePath, the DOR was 19.7734 (95% CI [1.6023; 244.0203]; I^2^ = 93.5%). The AUC was 0.841. Sensitivity, specificity, negative predictive value, and positive predictive value were 0.8573 (95% CI [0.6806; 0.9442]; I^2^ = 86.4%), 0.8368 (95% CI [0.4211; 0.9731]; I^2^ = 94.6%), 0.7573 (95% CI [0.5036; 0.9057]; I^2^ = 87.4%), and 0.8966 (95% CI [0.6509; 0.9758]; I^2^ = 92.1%), respectively.

There were no statistically significant differences in diagnostic accuracy between CS, ThinPrep, and SurePath (Appendix A). However, when comparing the two kits (ThinPrep and SurePath), significant differences in the AUC (0.791 vs. 0.841; *p* = 0.019) were observed, suggesting that SurePath might be more accurate.

## 4. Discussion

FNAC of thyroid lesions is a classic, safe, and meaningful test, playing an important role in diagnosing thyroid lesions and guiding treatment [9]. However, the efficacy and superiority of CS and LBP for FNAC in diagnosing thyroid lesions remain contentious. We analyzed the diagnostic accuracy of CS and LBP by comparing DOR, sensitivity, specificity, and AUC.

Our findings revealed no significant difference in diagnostic accuracy between CS, LBP, and the combination of CS with LBP. The diagnostic accuracy of CS and LBP was similar, while the accuracy of combining CS with LBP was notably lower. Sensitivity, specificity, and negative predictive value were also lower when combining CS with LBP compared to CS and LBP individually, but the differences were not statistically significant. Previous studies have reported no significant difference in diagnostic accuracy between CS and LBP [10,30,40,50]. Sensitivity has varied, reported as 78.9–93.6% for CS and 65.9–93.9% for LBP [25,26,30].

In previous studies, combining CS with LBP has been shown to reduce unnecessary thyroidectomies. LBP serves as a useful adjunct diagnostic tool for CS to identify malignant or suspicious thyroid lesions [9,46]. The rate of non-diagnostic results decreased when CS was combined with LBP, although not significantly, compared to CS alone [51]. However, in our study, combining CS with LBP resulted in relatively low diagnostic accuracy. Rossi et al. noted that slide adequacy assessed via CS indicated an increase in the non-diagnostic rate when using CS and LBP together [52]. In LBP, cells are preserved in a solution, preventing the real-time determination of sample adequacy [23,31]. Nonetheless, LBP can result in a relatively lower non-diagnostic rate due to a clearer background and fewer drying artifacts, provided that CS is not employed for on-site adequacy evaluation [51]. If slide cellularity is insufficient, additional slides can be utilized [7,30]. The combination of CS and LBP enables efficient slide preparation without compromising slide adequacy [53,54].

Our subgroup analysis of LBP kits revealed no significant differences in diagnostic accuracy between CS, ThinPrep, and SurePath. While SurePath had a lower DOR, its AUC was significantly higher, and there were no significant differences in the DOR, sensitivity, specificity, negative predictive value, and positive predictive value. A previous study reported that SurePath or ThinPrep achieved similar or marginally improved sensitivity and specificity compared to CS [8]. However, most studies on SurePath have been conducted in Belgium or Korea, limiting their generalizability. Furthermore, other studies have reported high sensitivity and specificity for both CS and SurePath [55]. Further studies on different LBP kits and in various countries could enhance the generalizability of the results.

Regarding the RIS, previous studies have reported variability in LBP, ranging from 10% to 25% [24,38,56]. One study observed better sample adequacy in LBP compared to CS [8], while another indicated a higher inadequacy rate for LBP than CS [31]. Our findings are similar to a previous study that found that combining CS with LBP resulted in a lower RIS compared to using either method alone [5], although this was not statistically significant. In LBP, the RIS may increase due to cell dilution in suspension media or loss of colloids during processing [9]. Repeated processing using LBP could mitigate this, enhancing sample adequacy and diagnostic accuracy [56]. The accumulation of clinical data and a learning curve are essential to improve the adequacy of LBP samples. As a newer technology compared to CS, LBP requires enhanced technical skills, such as syringe cleaning, to address issues such as low levels of cytoplasm in samples. In addition, the learning curve for cytopathologists, particularly in recognizing colloids and follicles, must be improved [10]. It is also important to consider the potential unclear effects of preservative solutions and artifacts that reduce inflammatory cells in LBP [10].

In previous studies, ease of interpretation has generally not correlated with the RIS. Only 3–5% of studies have evaluated LBP as being good for ease of interpretation compared to CS [9,38]. Moreover, studies that have assessed inadequate specimen rates have mainly focused on the SurePath or ThinPrep kits; more research is needed on other new LBP tests and technologies.

In our study, there were no significant differences between CS and LBP, and combining CS with LBP did not yield better results. Subgroup analyses also suggested that using CS or LBP alone might be preferable, with SurePath being the recommended choice when using LBP. However, the advantage of the combination could be considered in cases where specimen collection is challenging, as the RIS was lower, although not statistically significant.

Furthermore, cytomorphological differences between LBP and CS may vary in papillary, anaplastic, and medullary carcinoma. Papillary carcinoma can exhibit diverse cell arrangements [57,58]. Although CS and LBP do not allow for the detailed observation of tissue structure, understanding the clinical significance of morphological features is important [6]. Adding LBP can help distinguish between benign and malignant lesions due to better nuclear observation in a clear background [51]. Therefore, immunocytochemical and molecular studies should be concurrently considered for malignant or suspicious lesions as they can provide additional diagnostic assistance. If the FNAC results are uncertain, molecular testing for mutations such as BRAF and RAS can be useful [59]. Nuclei remain stable for up to 6 months in LBP preservative solution, potentially ensuring high reliability in mutation testing [7,60].

This study had several limitations. First, statistical heterogeneity was high, which is common in pathological studies [61], but the sampling methods and LBP techniques were not uniformly represented. The heterogeneity in the subgroup analyses for SurePath and ThinPrep might have stemmed from varied study designs and differences in LBP proficiency among examiners. The presence or absence of a cytopathologist and the use of different instruments and ultrasonography in the FNAC process also contributed to variability. In order to increase diagnostic accuracy, using the Bethesda Reporting System with ultrasonography, other appropriate diagnostic criteria could have been applied, such as Thyroid Imaging Reporting and Data System. Second, histologic follow-up was not included in the analysis, potentially limiting the evaluation, as most cases analyzed only initial diagnoses without considering final pathological diagnoses or modified FNAC diagnoses post-surgery. Further studies that incorporate histological follow-up are necessary. Third, the retrospective nature of several studies could have introduced bias, as non-diagnostic nodules were excluded after surgery. Cases suspected of follicular neoplasm or those without surgical intervention for non-diagnostic lesions may have been omitted. Fourth, CS, LBP, and their combination might not have been performed on the same thyroid lesion. Fifth, because the cytologic categories were different for each included study, it was difficult to evaluate benign and atypia lesions. Further studies evaluating malignancy risk for benign and atypia lesions in included studies with the same cytologic category are needed. Finally, most studies were from the United States, Europe, and Korea, possibly introducing bias due to limited racial diversity.

To supplement the accuracy and feasibility of CS and LBP, repeated processing is required, along with improving sample adequacy and diagnostic accuracy. The accumulation of clinical data and a learning curve are critical for improving the adequacy of LBP samples. LBP necessitates more advanced technical skills, and the learning curve for cytopathologists needs to be improved. Our subgroup analyses indicated that using CS or LBP alone may be preferable, with SurePath being the recommended option when using LBP. Although not statistically significant, the combination’s benefit might be taken into account in situations where collecting specimens is difficult.

## 5. Conclusions

There were no significant differences in diagnostic accuracy and RIS among CS, LBP, and their combination. While combining CS and LBP resulted in lower diagnostic accuracy and a decreased RIS, CS and LBP demonstrated similar accuracy. There were no significant differences in diagnostic accuracy among CS, ThinPrep, and SurePath. However, significant differences in the AUC suggest that the SurePath kit might be more accurate. Therefore, when choosing FNAC methods, cost, feasibility, and accuracy should all be considered.

## Figures and Tables

**Figure 1 cancers-16-00751-f001:**
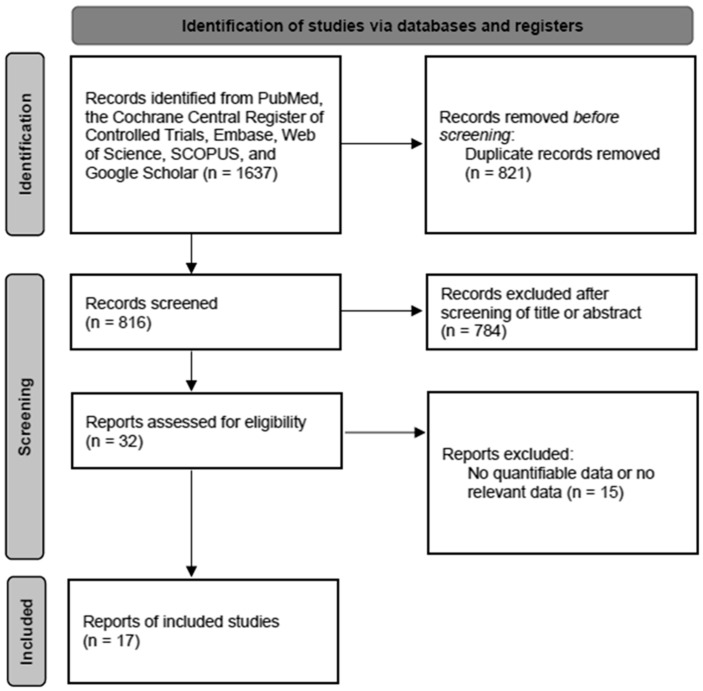
Diagram of the selection of studies for meta-analysis.

**Figure 2 cancers-16-00751-f002:**
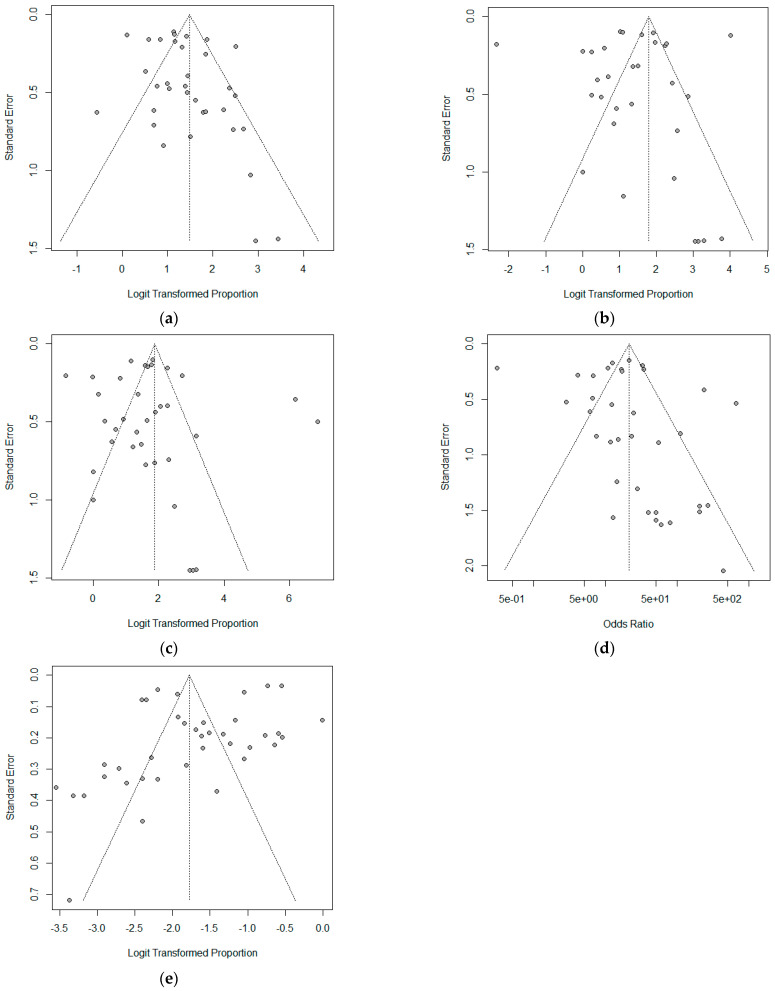
Begg’s funnel plot. (**a**) sensitivity, (**b**) specificity, (**c**) negative predictive value, (**d**) diagnostic odd ratios, and (**e**) rate of inadequate specimens.

**Figure 3 cancers-16-00751-f003:**
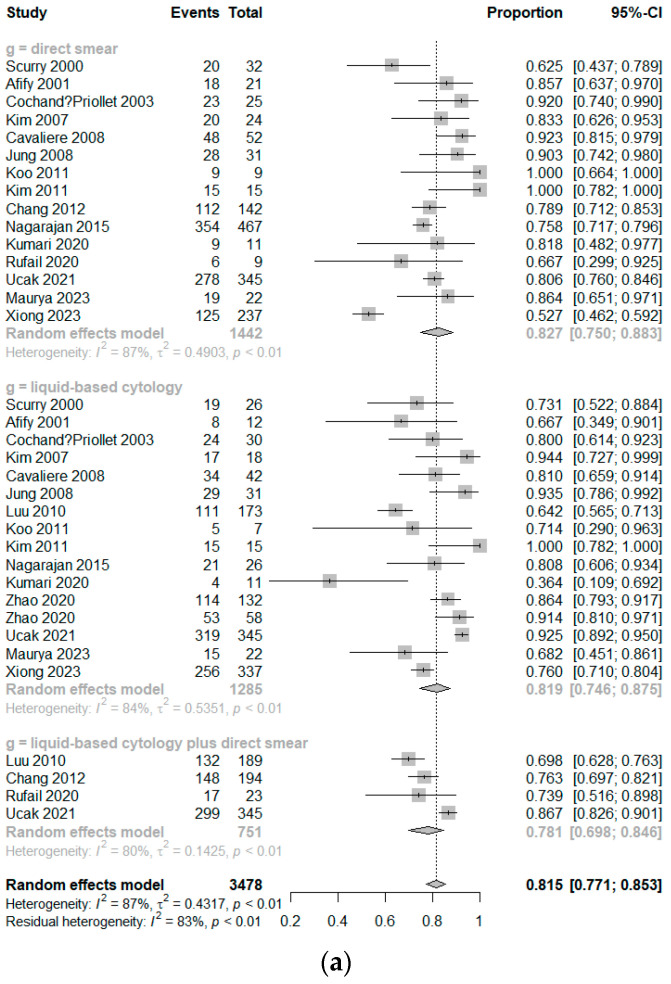
Forest plots of sensitivities (**a**), specificities (**b**), negative predictive values (**c**), diagnostic odd ratios (**d**), and rate of inadequate specimen (**e**).

**Figure 4 cancers-16-00751-f004:**
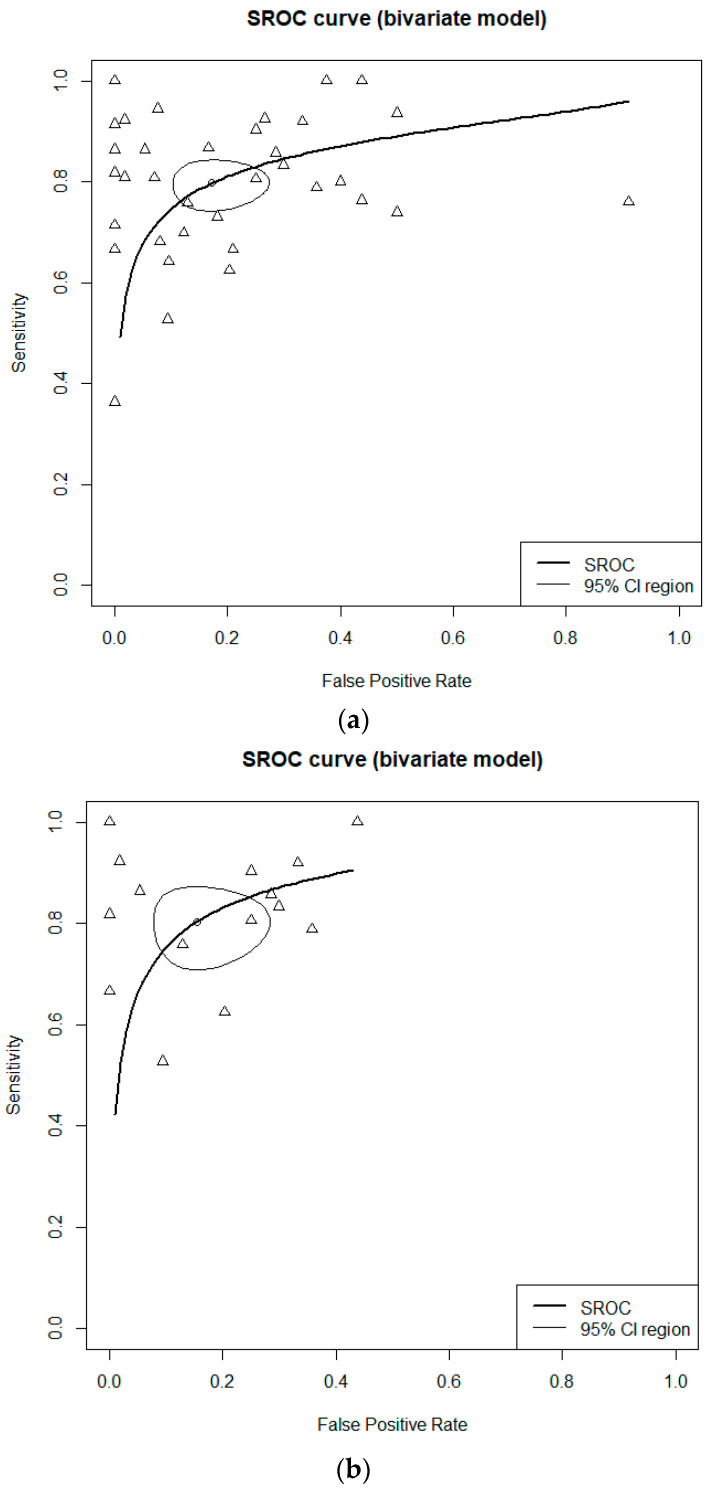
The summary receiver operating characteristic curve of (**a**) all included studies, (**b**) conventional smear, (**c**) liquid-based preparations, and (**d**) combination of conventional smear with liquid-based preparations. Thick curve line (summary receiver operating characteristic curve), thin circular line (95% confident region), and small circle (summary estimate).

**Table 1 cancers-16-00751-t001:** The characteristics of the included studies.

Study	Design	Total Number of Patients (n)	Age of Patients (Years, Median (Range) or Mean ± SD)	Sex (F/M)	Nationality	Comparison	Inadequate Smear (n/N)	Reference Test
Scurry 2000 [22]	Retrospective	327	48 (19–84)	189/138	Canada	Direct smear	40/109	Surgery confirmed
Scurry 2000 [22]	Retrospective	327	48 (19–84)	189/138	Canada	Thin-Prep liquid-based preparation	31/90	Surgery confirmed
Afify 2001 [23]	Retrospective	209	NA	NA	USA	Thin-Prep liquid-based preparation	26/95	Surgery confirmed
Afify 2001 [23]	Retrospective	209	NA	NA	USA	Thin-Prep liquid-based preparation	26/95	Surgery confirmed
Afify 2001 [23]	Retrospective	209	NA	NA	USA	Direct smear	9/46	Surgery confirmed
Afify 2001 [23]	Retrospective	209	NA	NA	USA	Direct smear	9/46	Surgery confirmed
Cochand-Priollet 2003 [24]	Case control	50	46.9 (24–70)	36/14	France	Direct smear	10/120	Surgery confirmed
Cochand-Priollet 2003 [24]	Case control	50	46.9 (24–70)	36/14	France	Thin-Prep liquid-based preparation	27/120	Surgery confirmed
Malle 2006 [17]	Retrospective	459	NA	NA	Greece	Direct smear	8/285	Only inadequate smear checked for this study
Malle 2006 [17]	Retrospective	459	NA	NA	Greece	Thin-Prep liquid-based preparation	7/174	Only inadequate smear checked for this study
Kim 2007 [41]	Prospective	172	NA	NA	Korea	SurePathTM liquid-based cytology	16/172	Surgery confirmed
Kim 2007 [41]	Prospective	172	NA	NA	Korea	Direct smear	36/172	Surgery confirmed
Stamataki 2007 [45]	Retrospective	157	NA	NA	Greece	Thin-Prep liquid-based preparation		Surgery confirmed
Cavaliere 2008 [25]	Prospective	3875	NA	NA	Italy	Thin-Prep liquid-based preparation	1252/3875	Surgery confirmed
Cavaliere 2008 [25]	Prospective	3875	NA	NA	Italy	Direct smear	1416/3875	Surgery confirmed
Jung 2008 [26]	Prospective	193	48.9 (20–81)	164/29	Korea	Direct smear	12/193	Surgery confirmed
Jung 2008 [26]	Prospective	193	48.9 (20–81)	164/29	Korea	SurePathTM liquid-based cytology	10/193	Surgery confirmed
Saleh 2008 [36]	Retrospective	126	NA	NA	USA	Direct smear	45/126	Only inadequate smear checked for this study
Saleh 2008 [36]	Retrospective	126	NA	NA	USA	Thin-Prep liquid-based preparation	40/126	Only inadequate smear checked for this study
Ardito 2010 [46]	Retrospective	353	45.8 (13–82)	267/86	Italy	Thin-Prep liquid-based preparation + Direct smear		Surgery confirmed
Geers 2010 [47]	Retrospective	178	NA	NA	Belgium	SurePath liquid-based preparation		Surgery confirmed
Luu 2010 [27]	Prospective	4101	NA	NA	USA	Thin-Prep liquid-based preparation + Direct smear	174/2000	Surgery confirmed
Luu 2010 [27]	Prospective	4101	NA	NA	USA	Thin-Prep liquid-based preparation	173/2102	Surgery confirmed
Koo 2011 [28]	Prospective	30	NA	NA	Korea	CellprepPlus liquid-based preparation	96/193	Surgery confirmed
Koo 2011 [28]	Prospective	30	NA	NA	Korea	Direct smear	32/193	Surgery confirmed
Kim 2011 [29]	Prospective	30	NA	NA	Korea	Direct smear		Surgery confirmed
Kim 2011 [29]	Prospective	30	NA	NA	Korea	SurePath liquid-based preparation		Surgery confirmed
Chang 2012 [30]	Prospective	4290	50	3662/628	Korea	Direct smear	458/1767	Surgery confirmed
Chang 2012 [30]	Prospective	4290	50	3662/628	Korea	Thin-Prep liquid-based preparation + Direct smear	318/2523	Surgery confirmed
Nagarajan 2015 [31]	Retrospective	1407	50 (7–84)	NA	USA	Direct smear	516/5169	Surgery confirmed
Nagarajan 2015 [31]	Retrospective	1407	50 (7–84)	NA	USA	Liquid-based preparation (not specified)	52/306	Surgery confirmed
Chang 2016 [48]	Retrospective	30	NA	NA	Korea	EASYPREPV liquid-based preparation	21/253	Surgery confirmed
Chang 2016 [48]	Retrospective	30	NA	NA	Korea	SurePath liquid-based preparation	15/253	Surgery confirmed
Gupta 2018 [37]	Prospective	60	NA	NA	India	Thin-Prep liquid-based preparation	2/60	Surgery confirmed
Gupta 2018 [37]	Prospective	60	NA	NA	India	Direct smear	5/60	Surgery confirmed
Kumari 2020 [32]	Prospective	100	NA	NA	India	Direct smear	10/100	Surgery confirmed
Kumari 2020 [32]	Prospective	100	NA	NA	India	SurePath liquid-based preparation	14/100	Surgery confirmed
Rufail 2020 [33]	Retrospective	584	NA	NA	USA	Direct smear	19/73	Surgery confirmed
Rufail 2020 [33]	Retrospective	584	NA	NA	USA	Thin-Prep liquid-based preparation + Direct smear	65/511	Surgery confirmed
Zhao 2020 [34]	Retrospective	221	20–76	178/43	China	Thin-Prep liquid-based preparation		Surgery confirmed
Zhao 2020 [34]	Retrospective	221	20–76	178/43	China	SurePath liquid-based preparation		Surgery confirmed
Mahajan 2021 [38]	Prospective case–control	200	21–72	170/30	India	Direct smear	7/200	Surgery confirmed
Mahajan 2021 [38]	Prospective case–control	200	21–72	170/30	India	SurePath liquid-based preparation	36/200	Surgery confirmed
Ucak 2021 [35]	Retrospective	879	46.7 (18–82)	700/179	Turkey	Liquid-based preparation (not specified)		Surgery confirmed
Ucak 2021 [35]	Retrospective	879	46.7 (18–82)	700/179	Turkey	Direct smear		Surgery confirmed
Ucak 2021 [35]	Retrospective	879	46.7 (18–82)	700/179	Turkey	Liquid-based preparation (not specified) + Direct smear		Surgery confirmed
Alam 2022 [39]	Prospective	131	33.15 ± 12.38	NA	India	SurePath liquid-based preparation	22/131	Surgery confirmed
Alam 2022 [39]	Prospective	131	33.15 ± 12.38	NA	India	Direct smear	9/131	Surgery confirmed
Sayer 2022 [40]	Retrospective	572	54.3 ± 10.16	446/126	Turkey	Direct smear	63/266	Surgery confirmed
Sayer 2022 [40]	Retrospective	572	54.3 ± 10.16	446/126	Turkey	SurePath liquid-based preparation	49/359	Surgery confirmed
Erdoğan 2023 [49]	Retrospective	4855	41–60	4069/786	Turkey	SurePath liquid-based preparation	324/2898	Surgery confirmed
Erdoğan 2023 [49]	Retrospective	4855	41–60	4069/786	Turkey	Cytospin liquid-based cytology	250/957	Surgery confirmed
Maurya 2023 [9]	Prospective, observational	250	12–72	224/26	India	SurePath liquid-based preparation	39/250	Surgery confirmed
Maurya 2023 [9]	Prospective, observational	250	12–72	224/26	India	Direct smear	13/250	Surgery confirmed
Xiong 2023 [6]	Retrospective	337	21–71	266/71	China	SurePath liquid-based preparation		Surgery confirmed
Xiong 2023 [6]	Retrospective	337	21–71	266/71	China	Direct smear		Surgery confirmed

NA; not available.

**Table 2 cancers-16-00751-t002:** Individual non-randomized controlled trial methodological quality.

Study	Selection ^a^				Comparability ^b^		Exposure ^c^			The Newcastle–Ottawa Scale
	1	2	3	4	5A	5B	6	7	8	
Scurry 2000 [22]	Yes	Yes	Yes	Yes	No	No	Yes	Yes	Yes	7
Afify 2001 [23]	Yes	No	Yes	Yes	No	No	Yes	Yes	Yes	6
Cochand-Priollet 2003 [24]	Yes	Yes	Yes	Yes	No	No	Yes	Yes	Yes	7
Malle 2006 [17]	Yes	Yes	Yes	Yes	No	No	Yes	Yes	Yes	7
Kim 2007 [41]	Yes	No	Yes	Yes	No	No	Yes	Yes	Yes	6
Cavaliere 2008 [25]	Yes	Yes	Yes	Yes	No	No	Yes	Yes	Yes	7
Jung 2008 [26]	Yes	No	Yes	Yes	No	No	Yes	Yes	Yes	6
Saleh 2008 [36]	Yes	Yes	Yes	Yes	No	No	Yes	Yes	No	6
Luu 2010 [27]	Yes	Yes	Yes	Yes	No	No	Yes	Yes	Yes	7
Koo 2011 [28]	Yes	No	No	Yes	Yes	Yes	Yes	Yes	Yes	7
Kim 2011 [29]	Yes	Yes	Yes	Yes	No	No	Yes	Yes	No	6
Chang 2012 [30]	Yes	Yes	Yes	Yes	No	No	Yes	Yes	Yes	7
Nagarajan 2015 [31]	Yes	No	Yes	Yes	No	No	Yes	Yes	Yes	6
Gupta 2018 [37]	Yes	No	No	Yes	Yes	Yes	Yes	Yes	Yes	7
Kumari 2020 [32]	Yes	No	Yes	Yes	No	No	Yes	Yes	Yes	6
Rufail 2020 [33]	Yes	No	No	Yes	Yes	Yes	Yes	Yes	Yes	7
Zhao 2020 [34]	Yes	Yes	Yes	Yes	No	No	Yes	Yes	Yes	7
Ucak 2021 [35]	Yes	No	No	Yes	Yes	Yes	Yes	Yes	Yes	7
Mahajan 2021 [38]	Yes	Yes	Yes	Yes	No	No	Yes	Yes	Yes	7
Alam 2022 [39]	Yes	Yes	Yes	Yes	No	No	Yes	Yes	Yes	7
Sayer 2022 [40]	Yes	No	Yes	Yes	No	No	Yes	Yes	Yes	6
Maurya 2023 [9]	Yes	No	No	Yes	Yes	Yes	Yes	Yes	Yes	7
Xiong 2023 [6]	Yes	No	Yes	Yes	No	No	Yes	Yes	Yes	6

A star rating system was used to indicate the quality of a study, with a maximum of nine stars. A study could be awarded a maximum of one star for each numbered item within the selection and exposure categories. ^a^: Selection (4 items): adequacy of case definition; representativeness of the cases; selection of controls; and definition of controls. ^b^: Comparability (1 item): comparability of cases and controls on the basis of the design or analysis. ^c^: Exposure (3 items): ascertainment of exposure; same method of ascertainment for cases and controls; and non-response rate (same rate for both groups).

## Data Availability

No new data were created as part of this systematic review.

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
