# Peer review of "Comparison of Liquid-Based Preparations with Conventional Smears in Thyroid Fine-Needle Aspirates: A Systematic Review and Meta-Analysis"

_cancers, 2024, doi:10.3390/cancers16040751_

Round 1
Reviewer 1 Report
Comments and Suggestions for Authors
"Comparison of liquid-based preparations with conventional 2 smears in thyroid fine-needle aspirates: A systematic review 3 and meta-analysis" is a review article including a meta-analysis research about the diagnostic value of two cytological techniques applied to the thyroid cytology. The study is well written.
The following are my observations:
1. Introduction, lines 49-50: "it often yields uneven spreads or ambiguous results, primarily due to the presence of blood clots in thyroid tissue samples". It is a bit excessive to claim that traditional cytology often produces ambiguous results. I would propose to write that traditional cytology can give variable results depending also on the experience of the cytopathologist (Scappaticcio L et al. Repeat thyroid FNAC: Inter-observer agreement among high- and low-volume centers in Naples metropolitan area and correlation with the EU-TIRADS. Front Endocrinol (Lausanne). 2022;13:1001728. Published 2022 Sep 15. doi:10.3389/fendo.2022.1001728).
2. Methods and results: my main concerne is about the methods used to compare the diagnostic performance of the two cytological techniques. Thyroid cytology does not provide the dichotomous distinction between benign and malignant, but uses different diagnostic classes, each of which corresponds to an expected risk of malignancy (ROS).
- Did all the studies included in the meta-analysis use the same reporting system? If not, the authors should specify which reporting systems were used in the different studies and how the results of different reporting systems were compared.
- Do the authors consider only the class of positive for malignant cells, or have they considered all diagnostic classes? If so, how sensitivity was calculated in the case of an intermediate diagnostic class (AUS-FLUS and others)?
- to compare the two cytological techniques, it would be necessary for the authors to compare the ROM in the different diagnostic classes in the traditional cytology group versus the liquid-based cytology group. In thyroid cytology, it is not only important to make a diagnosis of malignancy when sampling a papillary carcinoma, but it is necessary to respect the ROM expected in the different diagnostic classes. Which cytological technique is the best from this point of view?
Comments on the Quality of English LanguageMinor editing of English language required
Reviewer 2 Report
Comments and Suggestions for Authors
The article titled "Comparison of Liquid-Based Preparations with Conventional Smears in Thyroid Fine-Needle Aspirates: A Systematic Review and Meta-Analysis" presents a comprehensive evaluation of conventional smears (CS) and liquid-based preparations (LBP) in diagnosing thyroid malignant or suspicious lesions using fine-needle aspiration cytology (FNAC). The study involved a systematic review and meta-analysis of 17 studies, encompassing 15,861 samples, and compared the diagnostic accuracy, rate of inadequate smears, and subgroup analyses between CS, LBP, and two common LBP kits, ThinPrep and SurePath​​​​​​​​.
Scientific English Accuracy:
The language used in the article is appropriate for scientific English. The terminology, structure, and phrasing are consistent with academic writing standards.
Evaluation of Results:
The results are presented clearly, with appropriate statistical measures such as diagnostic odds ratio (DOR), area under the curve (AUC), sensitivity, specificity, negative predictive value, and positive predictive value.
The article reports no significant difference in diagnostic accuracy or the proportion of inadequate smears between CS and LBP. It also notes that SurePath demonstrated higher diagnostic accuracy than ThinPrep.
The statistical methods are well-documented and appropriate for the analysis conducted.
Conceptual Criticisms in Methods and Analysis:
The literature search strategy and selection criteria are clearly defined, ensuring the relevance and specificity of the included studies.
However, it is not clear which reporting system was used for thyroid cytology. If, as I believe, the Bethesda System was used, it would be necessary to specify how Bethesda III and IV results were interpreted. It should be clarified whether the articles included in the meta-analysis evaluated such results, and if so, how they did so, and how many of these results were present in each article.
Furthermore, in the discussion, it would be useful to mention the improvement in diagnostic accuracy through the association of the Bethesda Reporting System with ultrasound, applied with appropriate diagnostic criteria (TIRADS, ACE-AACE-AME, or others)
Reviewer 3 Report
Comments and Suggestions for Authors
Kang et al. compared the diagnostic accuracy and RIS of conventional smears (CS) and lipid-based preparation (LBP) methods of TN specimens taken by fine needle aspiration cytology (FNAC). The paper is well-written, systematic, actual, and important for everyday clinical practice. The introduction section is informative and emphasizes the study's rationale. The statistical methods are appropriate. The results are understandable and correctly presented. The discussion is fruitful and presented in a logical manner, as are the limitations. The references are updated and appropriate. But what the authors must improve are the usefulness of the presentation and the future perspectives. The authors should write several lines about the hints that will complement the accuracy and feasibility of both methods assessed as very similar regarding diagnostic accuracy and RIS (among CS, LBP, and their combination).
Reviewer 4 Report
Comments and Suggestions for Authors
I have read the sent article very carefully and I consider that the subject is of interest not only from the point of view of the accuracy of a histopathological diagnosis but also from the new perspective of determining the genetic alteration associated with some personalized therapies.
I believe that the authors have demonstrated high research qualities by being very careful in the elaboration of this review. They presented the results obtained, but also pointed out the possible bias.
Comments on the Quality of English Languageminor
Round 2
Reviewer 1 Report
Comments and Suggestions for Authors
The Authors complied with my previous requests.
Reviewer 2 Report
Comments and Suggestions for Authors
Following the review process, the manuscript has achieved high quality. The article has significant clinical importance and adds elements of interest to the current knowledge. Therefore, I consider the article suitable for publication in Cancers.